# MaestraNatura Reveals Its Effectiveness in Acquiring Nutritional Knowledge and Skills: Bridging the Gap between Girls and Boys from Primary School

**DOI:** 10.3390/nu15061357

**Published:** 2023-03-10

**Authors:** Rosaria Varì, Annalisa Silenzi, Antonio d’Amore, Alice Catena, Roberta Masella, Beatrice Scazzocchio

**Affiliations:** 1Center for Gender-Specific Medicine, Istituto Superiore di Sanità, Viale Regina Elena 299, 00161 Rome, Italy; 2Postgraduate School of Hygiene and Preventive Medicine, University of Brescia, 25123 Brescia, Italy

**Keywords:** health lifestyles, nutrition, education, students

## Abstract

MaestraNatura (MN) is a nutrition education program developed to both enhance awareness about the importance of healthy eating behaviour and skills on food and nutrition in primary school students. The level of knowledge about food and nutritional issues was assessed by a questionnaire administered to 256 students (9–10 years old) attending the last class of primary school and was compared with that of a control group of 98 students frequenting the same schools that received traditional nutrition education based on curricular science lessons and one frontal lesson conducted by an expert nutritionist. The results indicated that students in the MN program showed a higher percentage of correct responses to the questionnaire when compared with the control group (76 ± 15.4 vs. 59 ± 17.7; *p* < 0.001). Furthermore, the students attending the MN program were requested to organise a weekly menu before (T0) and when finished (T1) the MN program. The results evidenced an overall significant improvement in the score obtained at T1 with respect to those at T0 (*p* < 0.001), indicating a strong improvement in the ability to translate the theoretical concepts of nutrition guidelines in practice. In addition, the analysis revealed a gender gap between boys and girls, with boys showing a worse score at T0 that was ameliorated after the completion of the program (*p* < 0.001). Overall, MN program is effective in improving nutrition knowledge amongst 9–10-year-old students. Furthermore, students showed an increased ability to organise a weekly dietary plan after completing the MN program, a result which also bridged gender gaps. Thus, preventive nutrition education strategies specifically addressed to boys and girls, and involving both the school and family, are needed to make children aware of the importance of a healthy lifestyle and to correct inadequate eating habits.

## 1. Introduction

Childhood and adolescence are crucial times to lay the foundations of health in adult life; in particular, childhood is a period of biological and social change [1] in which eating behaviour often becomes unhealthy [2]. Excess weight in this phase of life represents an important challenge for global public health [3]. Childhood obesity, indeed, increases the risks of obesity in adult life [4] and those of developing type 2 diabetes, hypertension, dyslipidaemia [5], and CHD [6,7,8]. In this perspective, nutrition knowledge is recognised as an important factor able to promote healthy food choices, reversing the rise in the prevalence of overweight and obesity and their clinical consequences. However, numerous studies carried out mainly in adult people analysed the factors that influence the selection and consumption of food, highlighting that in Western societies sex-related biological or psychological factors as well as gender-related socioeconomic and cultural aspects can strongly influence food choices [9,10].

Women tend to select healthier food and are much more concerned than men with choosing appropriate food and healthy nutritional behaviours important for maintaining a good physical condition [9,10]. Thus, men should benefit from specific nutrition interventions considering the gender gap, which puts them at a disadvantage [10,11]. The urgent need to prevent or reverse the course of childhood obesity has led to the significant growth in research addressed at designing interventions to favour health-promoting dietary choices and to discourage those adversely affecting health.

In this regard, increasing knowledge of food and nutritional issues appears as a necessary basis for the improvement of eating behaviours [9,11]. Due to the reciprocal relationship between health and education, schools are among the most effective settings for health promotion in terms of influencing the adolescents’ dietary habits [12,13]. Ideally, a school-based intervention should be carried out during school curriculum hours or an after-school program in order to reduce the attrition of participants [14]. In addition, in the school context, a holistic approach to health promotion could be established, allowing the involvement of families and communities to reinforce health messages outside the school environment [15]. Various interventions have been carried out so far in the school setting; however, only few of them have evaluated the real effectiveness in improving nutrition knowledge and dietary habits among children [14,16]. Furthermore, most of the interventions consisted of few and short lessons which were not organised in a structured education program and generally without planning practical activities or the involvement of parents [17,18]. The MaestraNatura Program (MNP) is an innovative nutrition education program developed by Istituto Superiore di Sanità to guide teachers, parents, and children attending primary and first-level secondary schools through an 8-year learning path [19,20].

The program has several innovative points. First of all, the didactic activities span the entire scholastic year and require the active participation of students in experimental activities and the involvement of parents in practical applications. Then, the understanding of the principles of dietary guidelines for healthy eating represents the endpoint, instead of the starting point, to be reached after gaining knowledge about nutritional facts.

Finally, the MNP takes advantage of a web platform that makes it potentially spread and easily adapted everywhere, allowing the standardisation of the intervention [19,20].

The present study was aimed at investigating whether the MNP didactic path specifically developed for the students in their last class of primary school (9–10 years old) was able to favour the increase of knowledge about food and nutrition and related skills in young children.

## 2. Materials and Methods

### 2.1. Participants

Twenty fifth-year classes of primary public schools were enrolled in the study. The schools were located in the north (5 classes), centre (9 classes), and south (6 classes) of Italy, in both small and large towns. A total number of 354 students (181 girls, 173 boys), aged 9–10 years, participated in the study. In each school, a MN group and a CO group were organised so that the socio-cultural and economic characteristics of the students were as homogeneous as possible (Figure 1).

### 2.2. Ethical Aspects

Parents signed the informed consent to allow the participation of their children in the MNP as required by the Italian law regarding ethical and legal (personal data protection) aspects. The objectives of the study and the required activities were explained to teachers and parents in meetings and leaflets before the start of the didactic activities. The study was approved by the ethics committee of Istituto Superiore di Sanità (AOO-ISS 26.04.21 n.0015951) [20].

### 2.3. Procedure

The CO group attended curricular lessons (two 1 h lessons) about scientific topics related to nutrition and one frontal lesson (2 h) conducted by an expert nutritionist which focused on food groups, different meanings of food and nutrients, and the food pyramid [21]. The MN group took part in all the theoretical and practical activities planned by the MN educational path “Why do we have to eat?” that included three PowerPoint presentations (“Why do we have to eat?”, “Discover the egg”, and “Discover the milk”) and several experiments aimed at increasing nutrition knowledge, thus promoting awareness of the importance of a balanced and varied diet (“What’s in egg?”, “What’s in milk?”, “What food group does it belong to?”, “Plan a weekly menu” [19]). In addition, the learning path included a “how to cook” section, reporting recipes to cook at home together with parents in order to promote interaction between them and encourage children to taste new food, especially vegetables. The MNP didactic activities spanned the entire school year. It was possible to download all the contents from the MN web platform, which is divided into different areas specifically addressed to teachers, parents, and students. To compare the level of knowledge obtained through the two different learning approaches, CO and MN groups were required to fill in the multiple-choice questionnaire “What do you know about food and nutrition?” within one week from the end of the activities.

Furthermore, each student from the MN group was asked to compile a weekly food plan (WFP), both before starting any activity related to nutrition issues (T0) and at the end of the didactic path (T1). In this way, each student had their own starting level (T0) and any improvements due to participation in the program were compared to the basic level. The task consisted of the construction of a weekly menu that included breakfast, a morning snack, lunch, an afternoon snack, and dinner for each day of the week. The total score was calculated by counting the number of breakfasts, servings of fruit, vegetables, fish, cereals, and legumes. From this score, points were subtracted for the incorrect use of protein-rich food. By comparing the scores totalised at T0 and T1 from each student it was possible to assess the improvement, if there was any, in children’s performance in terms of translating their acquired knowledge into the arrangements of daily meals.

### 2.4. Statistical Analysis

Quantitative variables were analysed by means, standard deviation, medians, and ranges, while categorical variables by absolute and percent frequencies. With respect to the questionnaire, for any item the answer given by a single child was categorised as correct (1 out of 4 possible answers) or incorrect (3 out of 4 possible answers). Considering the proportion of children giving the correct answers, for any single item the difference between CO and MN groups was assessed by the Fisher’s exact probability test because of the expected presence of low frequencies in many items. The difference between the CO and MN groups was significant when the Fisher’s exact probability test was *p* < 0.05. In addition, for all children, we computed the average proportion of items receiving correct answers on the whole questionnaire, that is, a sort of global correctness index (GCI) of answers to the questionnaire. Differences between the two groups with respect to the GCI were assessed by Student’s *t* test. To compare the WFP organised by the MN group at the beginning and at the end of the didactic activities, the scores obtained were analysed by Student’s *t* test. *p* values < 0.05 were considered significant. All statistical analyses were performed by STATA 16.0.

## 3. Results

### 3.1. Evaluation of the Improvement in Knowledge Obtained by the “Why Do We Have to Eat?” Path with Respect to a ‘Traditional’ Nutrition Education Intervention

The questionnaires filled by two hundred and forty-three students (corresponding to about the 80% of the total students enrolled) were collected and analysed. The data revealed that MN group showed better performance in answering the administered questions compared with the CO group (*p* < 0.001); differences between boys and girls were not evident. On average, the percentage of correct answers was significantly higher in the MN group compared with the CO group (mean = 76.21 SD = 15.36 vs. mean = 59.25, SD = 17.68; *p* < 0.001) (Figure 2).

MN group students were able to answer most of the items correctly, showing a significant difference compared with the CO group (Table 1). It is worth noting that when it was not possible to observe significant differences between the groups, both of them answered correctly, probably because the questions dealt with everyday life concepts with which the students were already familiar.

### 3.2. Weekly Food Plan Organisation

The students, after finishing the MN program, showed an improved ability in planning their weekly menus. The scores obtained by the students at T1, indeed, were significantly higher with respect to those at T0 (*p* < 0.001) (Table 2).

By analysing the individual scores obtained for the servings of fruit, vegetables, and fish added to the plan separately, we found that all the students had significant increases in these categories at T1 (*p* < 0.001, *p* < 0.001, *p* = 0.047, respectively), serving as evidence of their understanding and learning the right consumption frequencies for those foods reported by the guidelines for healthy nutrition (Table 2). By considering the data distribution in urban areas, the most evident effect of the efficacy of the MNP was found in small towns (Table 2). It is worth noting that students from large cities exhibited a starting level significantly higher than those from small ones (*p* < 0.001), but this difference was largely resolved and even completely removed at the end of the didactic path (*p* < 0.001). The data indicated that students from the small towns significantly improved their skills by planning the right number of servings of fruit, vegetables, fish, cereals and legumes in all parts of their weekly menus (Table 2). Moreover, the data disaggregated for sex showed that boys started from a lower basic level of knowledge at T0 with respect to the girls (*p* = 0.029) (Table 3), but they reached a remarkable significant improvement at the end of the path, even if the difference with the girls remained statistically significant (*p* = 0.018) (Table 3). Of note, the differences observed between boys and girls were also found in the small towns but disappeared in the large ones.

## 4. Discussion

The present study showed the effectiveness of the innovative nutrition education intervention program MaestraNatura to increase knowledge and skills about food and nutrition in a sample of students, 9–10 years old, from the fifth class of primary school. Nutritional knowledge has been suggested to possibly play a role in the adoption of healthier food habits [22,23]. Healthy lifestyles are fundamental to fight obesity and overweight, which are principal challenges for the healthcare systems and serve as the main risk factors for the onset of non-communicable diseases [3,24].

The improvement of knowledge is considered a necessary step to increase the population’s health literacy and, especially, food literacy, being proposed as effective tools in improving the adherence to adequate dietary habits [25], as also indicated in the “White Paper on nutrition-, overweight-, and obesity-related health issues”, published in 2008 by the European Commission [26]. However, increasing knowledge, although necessary, might not be enough to promote effective and long-lasting effects on dietary habits without a concomitant acquisition of skills that enable people to translate the theory in practice, i.e., to implement the food pyramid principles into their daily diets [20,27,28]. This aspect was highlighted in a recent systematic review that pointed out how nutrition education interventions based on practical activities are more likely to be successful in improving diet quality among children [29].

Therefore, a meaningful way to improve health literacy and prevent improper eating behaviour might be to design effective educational intervention programs aimed at ameliorating not only knowledge, but especially attitude and practices related to food and nutrition in primary school [28].

Our results showed that the primary school students who underwent the nutrition education program MNP had more knowledge about nutrition, as assessed by the knowledge questionnaire, than those attending traditional programs, as was previously proven in middle school students [20].

It is interesting to note that these young students from primary school achieved a rate of correct answers even better than that achieved by older students from middle school [20]. This clearly indicates that educational interventions should be performed as early as possible to provide better results and an effective improvement in food and nutrition knowledge. In fact, eating behaviours start to be acquired early in life and, when established, they are rarely changed; it is thus required to intervene as soon as possible to promote the adoption of healthy lifestyles and balanced diets [30].

As regards the improvement in awareness and skills regarding nutritional facts and correct eating habits, the MNP was proven to be able to improve the transfer of the recommendations reported in nutrition guidelines to practical scenarios, most likely also thanks to the practical activities the MNP proposed to the students.

In the present paper, the MNP showed its effectiveness on students that initially had worse nutrition-related scores in planning their weekly menus; these were mainly students from rural areas. However, after attending the MNP, they achieved a level of understanding comparable to that of students living in large cities. Interestingly, these findings are in line with previous studies on the influence of urbanisation on eating habits which indicate that city size is predictive of food behaviour [31,32]. This may happen because of a different level of interest and awareness about the relationship between diet and health and/or a stronger adherence to ‘traditional’ habits not always in line with healthy nutrition.

The young students’ demonstrated ability to organise their weekly food plans was greater with respect to the students from middle school; specifically, primary school students, although starting from a worse initial level, overturned this situation and greatly improved their ability at the end of the didactic path [20].

The limitations of this study are as follows. We lacked a control group to carry out the WFP task, so we were only able to show the improvement in skills within the MN group, and were missing knowledge assessments in the two groups of students before and after the specific interventions. More studies will be needed to substantiate the improvement in knowledge and skills conferred by the MNP.

Finally, it is worth noting that boys, regardless of the geographical areas considered, had a lower level of basic knowledge and less familiarity with nutrition than girls; however, they reached an adequate level of learning and scored good rankings in planning their weekly menus after the completion of the MN didactic activities. The different background and competence between the genders are in accordance with previous studies carried out in adult people [33] that showed a different attitude towards food and nutrition in men and women, highlighting that women have generally a more positive attitude towards healthy food than men, as well as a greater awareness about the influence of nutrition on health [34,35]. Our findings may suggest that sex and gender could influence dietary behaviour as early as childhood. This represents an interesting field of research that deserves further and deeper studies specifically addressed at identifying the main sex/gender-driven factors affecting food choices and consumption.

## 5. Conclusions

Overall, our data demonstrated that the MNP is effective in improving nutrition knowledge amongst 9–10-year-old students. Furthermore, students, after completing the MNP, showed an increased ability to organise a weekly dietary plan. However, additional studies with a control group are needed to substantiate the MNP’s effect on improving nutrition skills.

Our study also confirmed that practical activities, such as the experiments performed at school and the recipes cooked at home, are important determinants in increasing nutritional knowledge and awareness of healthy eating habits.

## Figures and Tables

**Figure 1 nutrients-15-01357-f001:**
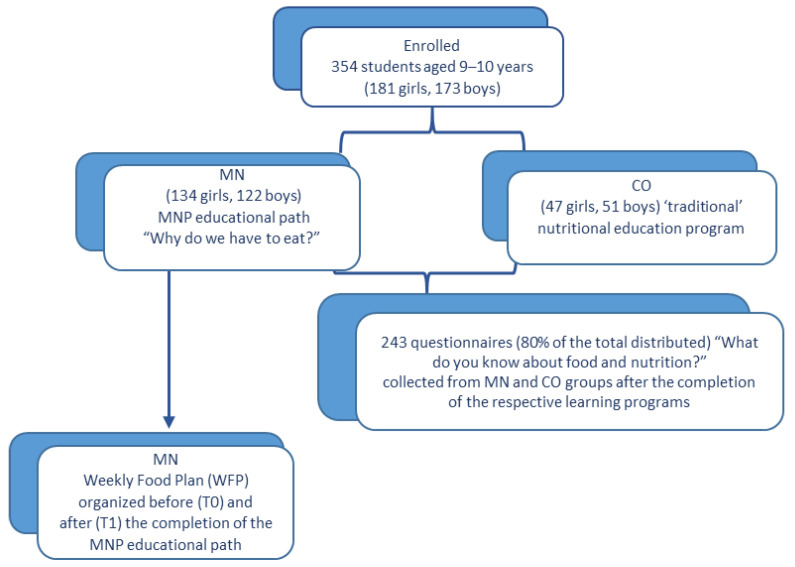
Flow chart of the study and subject recruitment. Students from MN and CO groups shared the same socio-cultural and economic context. MN: students attending the MNP educational path “Why do we have to eat?”; CO: students attending two 1 h lessons on nutrition-related topics conducted by their own teacher and one 2 h lesson conducted by an expert nutritionist.

**Figure 2 nutrients-15-01357-f002:**
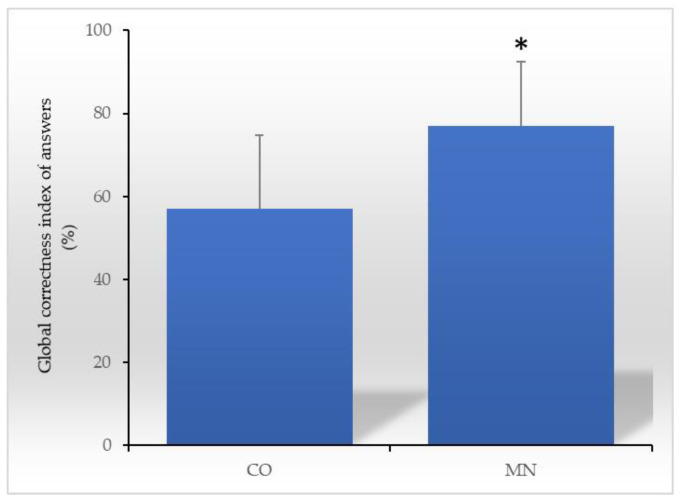
Average percentage of correct answers to the questionnaire “Why do we have to eat?”. Differences between the control group (CO) and the MaestraNatura (MN) group were calculated by Student’s *t* Test. * *p* < 0.001.

**Table 1 nutrients-15-01357-t001:** Percentage of students giving the correct answers to the questions in the questionnaire: “What do you know about food and nutrition?”.

Questions	% MN	% CO	*p*
What is digestion for?	59.86	50.00	0.155
2. What does water contain?	85.03	87.10	0.899
3. A portion of potatoes can replace…	74.15	33.87	<0.001
4. A portion of pasta and beans can replace…	88.44	33.87	<0.001
5. What are sugars for?	96.60	80.65	<0.001
6. Which of the following nutrients is a micronutrient?	74.83	67.74	0.227
7. The proteins are…	80.27	45.16	<0.001
8. Sugars can also be called…	82.99	33.87	<0.001
9. The fats can be…	78.23	70.97	0.195
10. What is the least important function of proteins?	65.99	20.97	<0.001
11. What are vitamins for?	57.14	56.45	0.832
12. Fats can also be called…	76.19	30.65	<0.001
13. Which of the following foods contains the least amount of water?	85.71	75.81	0.053
14. Which of the following foods does not contain fat?	75.51	90.32	0.028
15. Which of the following foods contains simple sugars?	75.51	61.29	0.026
16. Which of the following foods contains less saturated fat?	70.07	74.19	0.668
17. Do the cereals (wheat, barley, spelt, rye) contain proteins?	74.15	82.26	0.291
18. Which of the following foods contains more water?	54.42	58.06	0.718
19. Which of the following foods contains more fat?	78.91	59.68	0.002
20. Which of the following substances is not a nutrient?	53.74	32.26	0.003
21. Which of the following nutrients does not provide energy?	62.59	22.58	<0.001
22. What are the nutrients?	77.55	59.68	0.005
23. Which of the following substances is not a food?	95.24	88.71	0.041
24. Which of the following foods contains carbohydrates?	82.31	45.16	<0.001
25. Which of the following foods is not a cereal?	87.07	85.48	0.572
26. Which of the following foods is not a legume?	89.12	59.68	<0.001

Table 1 reports the percentage of students that correctly answered the single questions of the questionnaire. Differences between the control group (CO) and the MaestraNatura (MN) group were considered significant when *p* < 0.05.

**Table 2 nutrients-15-01357-t002:** Weekly food plan (WFP) evaluation.

	T0 Mean (SD)	T1 Mean (SD)	T1–T0	*p*
Total				
ALL	22.45 (7.8)	28.48 (9.8)	6.031	<0.001
Provinces	20.84 (7.43)	30.36 (9.85)	9.52	<0.001
Cities	25.61 (7.6)	24.6 (8.44)	−0.83	0.277
*p* (provinces vs. cities)	<0.001	<0.001		
Vegetables				
ALL	6.46 (4.25)	8.42 (4.32)	1.96	<0.001
Provinces	5.7 (4.15)	9.24 (4.11)	3.53	<0.001
Cities	7.95 (4.05)	6.74 (4.23)	−1.13	0.013
*p* (provinces vs. cities)	<0.001	<0.001		
Fruit				
ALL	8.08 (4.52)	11.73 (5.87)	3.64	<0.001
Provinces	7.43 (4.37)	12.87 (5.85)	5.44	<0.001
Cities	9.38 (4.5)	9.42 (5.22)	0.04	0.83
*p* (provinces vs. cities)	0.001	<0.001		
Fish				
ALL	0.77 (0.43)	0.84 (0.37)	0.070	0.047
Provinces	0.74 (0.44)	0.86 (0.34)	0.1	0.035
Cities	0.84 (0.04)	0.83 (0.37)	−0.023	0.685
*p* (provinces vs. cities)	0.071	0.536		
Cereals/legumes				
ALL	1.34 (1.24)	1.48 (1.22)	0.137	0.133
Provinces	1.25 (1.24)	1.5 (1.2)	0.25	0.037
Cities	1.51 (1.22)	1.39 (1.23)	−0.08	0.546
*p* (provinces vs. cities)	0.123	0.501		

Table 2 reports the mean ± SD of the score obtained in the WFP organised by the students. Differences between the score at T1 and T0 and between provinces and cities were calculated by Student’s *t* Test. *p* < 0.05 was considered significant.

**Table 3 nutrients-15-01357-t003:** Total and individual scores disaggregated by sex.

Total			
	T0 Mean (SD)	T1 Mean (SD)	*p* (T0 vs. T1)
F	23.47 (7.96)	32.12 (8.63)	<0.001
M	21.35 (7.51)	28.59 (10.7)	<0.001
*p* (F vs. M)	0.029	0.018	
Vegetables			
F	6.95 (4.12)	9.83 (3.53)	<0.001
M	5.93 (4.34)	8.63 (4.56)	<0.001
*p* (F vs. M)	0.056	0.022	
Fruit			
F	8.6 (4.64)	13.85 (5.44)	<0.001
M	7.54 (4.34)	11.88 (6.12)	<0.001
*p* (F vs. M)	0.057	0.043	
Fish			
F	0.79 (0.4)	0.86 (0.35)	0.074
M	0.75 (0.45)	0.80 (0.4)	0.298
*p* (F vs. M)	0.444	0.136	
Cereals/legumes			
F	1.42 (1.24)	1.6 (1.1)	0.555
M	1.26 (1.23)	1.4 (1.28)	0.119
*p* (F vs. M)	0.302	0.723	

The table reports the mean ± SD of the score obtained in the weekly meal plan organised by the students. Differences between the score at T1 and T0 and between males (M) and females (F) were calculated by Student’s *t* Test. *p* < 0.05 was considered significant.

## Data Availability

The data presented in this study are available on request from the corresponding author.

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
