# Peer review of "MaestraNatura Reveals Its Effectiveness in Acquiring Nutritional Knowledge and Skills: Bridging the Gap between Girls and Boys from Primary School"

_nutrients, 2023, doi:10.3390/nu15061357_

Round 1

Reviewer 1 Report

After reviewing the manuscript, which deals with an important topic, I conclude that it needs significant improvement before publication. The problem is methodological errors or failure to include important information in the article. For example, it is not clear what criteria were used to select the control group, why different tests were performed in the control and study groups. The results presented do not entitle one to conclude that MNP is more effective. The manuscript therefore needs to be reworked both in terms of content but also in terms of style and editing

Some detailed comments:

Introduction

The rationale for the choice of the research topic is quite obvious, due to the nutritional abnormalities in a group of children, mainly overweight and obesity.  Hence, there is no need to justify so extensively why nutrition education is so important. What is missing, however, is information about the educational programs used so far and the results achieved.  There is also no more detailed information about the Maestro Natura program, what makes it different, what knowledge (theoretical or practical) it has the potential to change. And information about the novelty of the study is also not indicated.

Materials and Methods

Why was the Weekly Food Plane before (TO) and after (T1) not used in the control group?

Was the level of knowledge assessed at the beginning, that is, before the start of nutrition education in both groups?

Lines 120 and 126: it would be good to write briefly about this in the methodology. Just citing the source is not enough to have a good understanding of the results. In addition, there would then be no need to include information on the method of counting under Table 3.

The characteristics of the study group are not presented.

Results

Lines 154 -156. "It is worth of note that, when it was not possible to observe significant differences between the groups, both of them answered correctly, probably because the questions dealt with everyday life concepts with which students were already familiar." If there had been a measurement of nutritional knowledge before nutrition education, there would have been no need to rely on such an interpretation of the results.

Table 2. under the table there is no explanation of what the different items are. It would be better to indicate the questions instead of the question numbers, and in this way Table 1 could be omitted (it lacks the proposed answers).

Line 171-174 Is this part of the table?

The description of the results needs to be improved, it should be more synthetic, leaving out a lot of wording, e.g. line 190 "The data clearly indicated, in fact, that".

Improving the tables is necessary. Once they are expanded on the page, they will fit on one page and it will be easier to evaluate the results as a whole.

Discussion

Lines 219-243. This is Introduction rather than Discussion. The study does not show what the study group looked like in terms of the prevalence of overweight and obesity.

I can't agree with the statement "Our results showed that the nutrition education MNP is more effective than the traditional interventions" if there was no measurement of pre-education knowledge in both groups. Instead, it can be said that the students participating in the MNP had more knowledge than the control group after the education was completed.

Author Response

Reviewer 1

After reviewing the manuscript, which deals with an important topic, I conclude that it needs significant improvement before publication. The problem is methodological errors or failure to include important information in the article. For example, it is not clear what criteria were used to select the control group, why different tests were performed in the control and study groups. The results presented do not entitle one to conclude that MNP is more effective. The manuscript therefore needs to be reworked both in terms of content but also in terms of style and editing

We warmly thank the reviewer for the suggestions. We modified the manuscript to address all the doubts and criticisms raised.

Some detailed comments:

Introduction

The rationale for the choice of the research topic is quite obvious, due to the nutritional abnormalities in a group of children, mainly overweight and obesity.  Hence, there is no need to justify so extensively why nutrition education is so important. What is missing, however, is information about the educational programs used so far and the results achieved.  

We accepted reviewer’s suggestions and modified the text accordingly (lines 35-51 and 53-68)

There is also no more detailed information about the Maestro Natura program, what makes it different, what knowledge (theoretical or practical) it has the potential to change. And information about the novelty of the study is also not indicated.

More detailed information on MaestraNatura program has been included, as well as about the novelty of the study (lines 69-78)

Materials and Methods

Why was the Weekly Food Plane before (TO) and after (T1) not used in the control group?

The weekly food plan was a specific activity of the MNP that allowed us to measure the improvement in the students’ skills only within the MN group. Each student from MN group was asked to compile a Weekly Food Plan, both before starting any activity related to nutrition issues (T0) and at the end of the didactic path (T1). In this way each student had his/her own starting level (T0) and any improvements due to the performing of the program was compared to the basic level. In addition, activities similar to the Weekly Food Plan are not typically carried out in the curricular program that represented the standard intervention for the control group. However, this represents a limitation of the study and other studies will be carried out to substantiate that MNP improved nutritional skills. All these considerations are now in the text (lines 97-99; 108-124).

Was the level of knowledge assessed at the beginning, that is, before the start of nutrition education in both groups?

We thank the reviewer for this questions that allowed us to improve and clarify this aspect. The design of the study was organized as a case-control study (MN, the case, and CO, the control group). To MN group was administered the MNP didactic path through the web platform www.maestranatura.org: power point presentation, exercises, evaluation tests, experiments, practical activities and recipes to make at school with teachers and at home with parents. The didactic intervention of MNP spanned the entire scholastic year; CO group got one frontal 2h-lesson by an expert nutritionist focused on food groups, different meaning of food and nutrients, and the significance of Food Pyramid plus curricular lessons (generally not more than two 1h-lesson) about scientific topics held by the teacher. The level of the acquired knowledge was measured and compared at the end of the respective interventions, within one week from the end of the activities (text modified accordingly, lines 97-99; 111-114)

Lines 120 and 126: it would be good to write briefly about this in the methodology. Just citing the source is not enough to have a good understanding of the results. In addition, there would then be no need to include information on the method of counting under Table 3.

The methodology was integrated and the information under table 3 was deleted as suggested.

The characteristics of the study group are not presented.

In each school, a MN group and a CO group were organised so that the socio-cultural and economic characteristics of the students of the two groups were as homogeneous as possible. We added a sentence to the text to clarify this point (lines 84-86).

Results

Lines 154 -156. "It is worth of note that, when it was not possible to observe significant differences between the groups, both of them answered correctly, probably because the questions dealt with everyday life concepts with which students were already familiar." If there had been a measurement of nutritional knowledge before nutrition education, there would have been no need to rely on such an interpretation of the results.

We thank the reviewer for the suggestion. However, as now stated in the materials and method section, the level of acquired knowledge was measured at the end of the respective interventions and compared between the groups.

Table 2. under the table there is no explanation of what the different items are. It would be better to indicate the questions instead of the question numbers, and in this way Table 1 could be omitted (it lacks the proposed answers).

Thank you for your suggestion; Table 1 and 2 were merged and table 1 was eliminated.

Line 171-174 Is this part of the table?

Yes, it is the footnote of the Table, that, however, was shortened as suggested by the reviewer 2

The description of the results needs to be improved, it should be more synthetic, leaving out a lot of wording, e.g. line 190 "The data clearly indicated, in fact, that".

As suggested by the reviewer, the description of the results was modified throughout the section.

Improving the tables is necessary. Once they are expanded on the page, they will fit on one page and it will be easier to evaluate the results as a whole.

As suggested by the reviewer, the tables were improved too; now they should fit on one page.

Discussion

Lines 219-243. This is Introduction rather than Discussion. The study does not show what the study group looked like in terms of the prevalence of overweight and obesity.

Those lines were eliminated in the Discussion.

I can't agree with the statement "Our results showed that the nutrition education MNP is more effective than the traditional interventions" if there was no measurement of pre-education knowledge in both groups. Instead, it can be said that the students participating in the MNP had more knowledge than the control group after the education was completed.

We modified the text according to the reviewer’s suggestions (lines 237-240)

Reviewer 2 Report

Please attached review notes.

Author Response

Reviewer 2 Notes

Title: MaestraNatura reveals its effectiveness in acquiring nutritional knowledge and skills bridging the gap between girls and boys from primary school

General comment:

- This paper presents the result of an intervention study that applied MaestraNatura Program (MNP)—specific nutrition education program designed for young students—in primary school setting in Italy. The findings show promise. However, the claims the authors made based on the intervention result did not seem entirely accurate (particularly the claim that the intervention improved nutrition skill). Also, the write up of the paper has a number of issues. Please specific comments below.

We warmly thank the reviewer for the comments that led us to improve the manuscript

Abstract

- Lines 24-28: reconsider revising this section in light my comments in the result and discussion/conclusion sections.

We revised the abstract according to the reviewer’s suggestions. Lines 24-26

Introduction

- Line 31-72—appears as one long paragraph- please break down, focusing on one key idea per paragraph.

The introduction was modified according to the reviewer’s suggestions (lines 35-51)

- Line 37-39: How does nutrition combat the rise of overweight and obesity…? Is it the nutrition or the healthier choices the individual makes around diet? Please qualify/clarify your statement. It’s not the diet but what the individual chooses to do with it that leads to a positive or negative outcome…

We apologize for the lacking of clarity in this sentence. Of course, the individual choices lead to correct/unhealthy eating habits, for this reason health literacy and, in particular, food literacy is determinant to help the individuals to make correct choices. The sentence was modified accordingly. (Lines 35-51)

- Line 41-46: Authors use the phrase “Numerous studies focused…” but cite just one source reference #10] which is an edited book with several chapters to support their argument (And source was not cited properly) …Only one section/chapter in the edited book discusses issues around gender and food choice. Please cite that specific chapter (see below), and either add other sources or remove the phare “numerous studies” from your sentence. [Claudia Arganini, Anna Saba, Raffaella Comitato, Fabio Virgili, Aida Turrini. Gender Differences in Food Choice and Dietary Intake in Modern Western Societies. In: Jay Maddock, editor. Public Health [Internet]. Rijeka: IntechOpen; 2012 [cited 2023 Feb 23]. p. Ch. 4. Available from: https://doi.org/10.5772/37886]

Thank you for the comment; we corrected the citation, added a new reference and modified the sentence

- Line 46: “considering all the above mentioned” what was mentioned above? Please be specific and state here. Also consider starting a new paragraph here

As suggested, the sentence was modified

- Line 52-54: “Boys and adult men in particular should be subjected to nutritional education, considering the gender gap which put them at a disadvantage [12].” The article cited here refers to grown up men and how their dietary lifestyle was distinct from women, men generally making less heathier choices. It does not comment about boys. Even in the former case, it is not clear if it is being male in and of itself versus factors associated with being a male that are influencing the nutrition decisions of the male gender. It would be good if authors clarify the distinction, also separating men vs. boys.

We modified the text according to the reviewer’s suggestion and added a new reference specifically addressed to boys eating behaviour.

- Line 65: please adequately describe the MNP, who developed, for what purpose and to be utilized where (in what context), etc.?

More detailed information on MaestraNatura program has been included, as well as the novelty of the study (lines 69-78)

Materials and methods

- Lines 94-111: can you clarify the total curriculum time for the MN and CO group? You mention 2hours for the frontal lesson by a nutrition expert for the CO group, but you do not mention the total time for the traditional vs the MN program.

We thank the reviewer for the question that allowed us to improve and clarify this aspect. The design of the study was organized as a case-control study (MN, the case, and CO, the control group). To MN group was administered the MNP didactic path through the web platform www.maestranatura.org: powerpoint presentation, exercises, evaluation tests, experiments, practical activities and recipes to make at school with teachers and at home with parents. The didactic intervention of MNP spanned the entire scholastic year; CO group got one frontal 2h-lesson by an expert nutritionist focused on food groups, different meaning of food and nutrients, and the significance of Food Pyramid plus curricular lessons (generally not more than two 1h-lesson) about scientific topics held by the teacher. The level of the acquired knowledge was measured at the end of the respective interventions, within one week from the end of the activities.

The text was modified accordingly (lines 97-99; 108-124)

- How soon after the completion of the program was the multiple-choice test was administered?

We apologize for the lacking of this information. The questionnaire was administered within one week from the end of the activities. This sentence is now present in the “Procedure” paragraph (lines 112-114)

- Line 113-121: What did the CO group do that was comparable to the Weekly Food Plan in the MN group? How could you attribute the improvement in score at T1 to the MN program if you did not have a control group for this activity?

The weekly food plan was a specific activity of the MNP that allowed us to measure the improvement in the students’ skills. Each student from MNP group was asked to compile a Weekly Food Plan, both before starting any activity related to nutrition issues (T0) and at the end of the didactic path (T1). In this way each student had his/her own starting level (T0) and any improvements due to the performing of the program was compared to the basic level. In addition, activities similar to the Weekly Food Plan are not typically carried out in the curricular program that represented the intervention for the control group. However, this represents a limitation of the study that is reported in the revised text (lines 264-266)

- Line 125-126 [and lines 120-121]- please briefly describe the evaluation methods here instead of asking the reader to find another article to understand the method…

The methods are now described in detail in the manuscript

- Lines 126-137 – describe the rationale for choosing to use Fisher’s exact test, Mann-Whitney U test, etc.

The rationale for choosing statistic tests is now described in the manuscript

Results

- Lines 144-147: You talk about comparing the % correct answer for the questionnaire for each group, but you provide mean and SD? Did you compare the average score of correct answer/questionnaire between the two groups or?

Yes, we compared the average score of correct answers between the two groups.

- Table 2: The numbers in the table do not look like they are percentages. If they are percentages (as pointed out at the table footnote), then why use Student’s t test? – not clear for this reviewer.

We apologize for the misunderstanding. The Table 2 was modified and merged with Table 1 in a new Table 1, also considering the suggestions of the Reviewer 1. There was a typo in the Table 2 that was amended. Data presented in the Table 2 are percentages and the test used for the statistic evaluation was Fisher’s and not Student’s.

- Double check the decimal places you can use when reporting p values (<0.001 or <0.0001).

Thank you for your comment, we checked the decimal places in the entire manuscript

- Table 3- reports just the within group comparison between T0 and T1 –no control group? How does this affect your result?

As said above, the weekly food plan was a specific activity of the MNP that allowed us to measure the score before (T0) and after (T1) a treatment (MNP activities) in order to evaluate the improvement in the students’ skills. For this reason a CO group was not provided for this specific activity.

Discussions

- Lines 212-214: The claim that MNP was effective in improving the knowledge and skills of 9–10- year-old learners is more than your data can support. If the skill piece is dependent on the Weekly Food Plan that the MN group did, it was only a within group comparison. You can describe what a happened but to attribute the improvement to the intervention with a control group may not be accurate. You would need to review your comments based on the improvement in WFP at T0 and T1 through out the paper.

We thank the reviewer for this comment that allow us to clarify this point. The WFP permitted to define for each subject the increase of his/her ability in planning a weekly menu, i.e. the capacity to perform a practical activity on the basis of the acquired knowledge. We think that it is possible to say that each student showed an improvement of such ability.

We cannot say that MNP is better than other intervention in doing this, because of the lack of a control group and from this point of view we totally agree with the reviewer.

We changed the text to better clarify this aspect.

- Lines 214-217: break down the sentence. The key message in the sentence is not clear…

The sentence was shortened.

- Discuss the limitation of your study.

The limitation of the study was added

Conclusion

- Your conclusion should highlight the major finding which is MNP being effective in improving nutrition knowledge amongst 9–10-year-old students… describe the improvement in the WFP for the MN group suggest additional study with a control group to substantiate MN indeed improves nutrition skills.

We changed the conclusion paragraph according to reviewer’s suggestions

Round 2

Reviewer 1 Report

Thank you for taking my suggestions into consideration. However, I still  suggest the improvements of the manuscript. Comparison of knowledge/skills between both groups (MN and CO) at the end stage of the study is of little value when the level of knowledge at the beginning is not known. Some specific comments are placed under your answers.

Some detailed comments:

The rationale for the choice of the research topic is quite obvious, due to the nutritional abnormalities in a group of children, mainly overweight and obesity. Hence, there is no need to justify so extensively why nutrition education is so important. What is missing, however, is information about the educational programs used so far and the results achieved.

We accepted reviewer’s suggestions and modified the text accordingly (lines 35-51 and 53-68)

I think that there is no need to discuss the motives of food choice among adults – lines 38-51.

There is also no more detailed information about the Maestro Natura program, what makes it different, what knowledge (theoretical or practical) it has the potential to change. And information about the novelty of the study is also not indicated.

More detailed information on MaestraNatura program has been included, as well as about the novelty of the study (lines 69-78)

 Please add the source of this information.

Why was the Weekly Food Plane before (TO) and after (T1) not used in the control group?

The weekly food plan was a specific activity of the MNP that allowed us to measure the improvement in the students’ skills only within the MN group. Each student from MN group was asked to compile a Weekly Food Plan, both before starting any activity related to nutrition issues (T0) and at the end of the didactic path (T1). In this way each student had his/her own starting level (T0) and any improvements due to the performing of the program was compared to the basic level. In addition, activities similar to the Weekly Food Plan are not typically carried out in the curricular program that represented the standard intervention for the control group. However, this represents a limitation of the study and other studies will be carried out to substantiate that MNP improved nutritional skills. All these considerations are now in the text (lines 97-99; 108-124).

In lines 97-99 there is information on the approval of the ethics committee

Was the level of knowledge assessed at the beginning, that is, before the start of nutrition education in both groups?

We thank the reviewer for this questions that allowed us to improve and clarify this aspect. The design of the study was organized as a case-control study (MN, the case, and CO, the control group). To MN group was administered the MNP didactic path through the web platform www.maestranatura.org: power point presentation, exercises, evaluation tests, experiments, practical activities and recipes to make at school with teachers and at home with parents. The didactic intervention of MNP spanned the entire scholastic year; CO group got one frontal 2h-lesson by an expert nutritionist focused on food groups, different meaning of food and nutrients, and the significance of Food Pyramid plus curricular lessons (generally not more than two 1h-lesson) about scientific topics held by the teacher. The level of the acquired knowledge was measured and compared at the end of the respective interventions, within one week from the end of the activities (text modified accordingly, lines 97-99; 111-114)

In lines 97-99 there is information on the approval of the ethics committee. In my opinion the groups a MN group and a CO group should have been organized according to the level of knowledge at the beginning of the study. Organizing them according the socio-cultural and economic characteristics  and lack of measurement of nutritional knowledge at the beginning  assumes that the knowledge of both groups  is the same. Such an assumption is wrong.  Comparison of knowledge/skills between both groups  at the end is of little value in such a situation.

The characteristics of the study group are not presented.

In each school, a MN group and a CO group were organised so that the socio-cultural and economic characteristics of the students of the two groups were as homogeneous as possible. We added a sentence to the text to clarify this point (lines 84-86).

The information “In each school, a MN group and a CO group were organized so that the socio-cultural and economic characteristics of the students were as homogeneous as possible” is not presented on Figure 1. The results are presented for about 80% of groups presented on Figure 1. This information should be added on Figure, especially when  the test was made only on the end stage of study.

Line 171-174 Is this part of the table?

Yes, it is the footnote of the Table, that, however, was shortened as suggested by the reviewer 2

This information is still too long

Author Response

Thank you for taking my suggestions into consideration. However, I still  suggest the improvements of the manuscript. Comparison of knowledge/skills between both groups (MN and CO) at the end stage of the study is of little value when the level of knowledge at the beginning is not known. Some specific comments are placed under your answers.

Some detailed comments:

The rationale for the choice of the research topic is quite obvious, due to the nutritional abnormalities in a group of children, mainly overweight and obesity. Hence, there is no need to justify so extensively why nutrition education is so important. What is missing, however, is information about the educational programs used so far and the results achieved.

We accepted reviewer’s suggestions and modified the text accordingly (lines 35-51 and 53-68)

I think that there is no need to discuss the motives of food choice among adults – lines 38-51.

We added these paragraphs because of the request by the other reviewer; thus, sorry but we cannot delete those lines

There is also no more detailed information about the Maestro Natura program, what makes it different, what knowledge (theoretical or practical) it has the potential to change. And information about the novelty of the study is also not indicated.

More detailed information on MaestraNatura program has been included, as well as about the novelty of the study (lines 69-78)

 Please add the source of this information.

We organized all the MaestraNatura contents at Istituto Superiore di Sanità; the references 19 and 20 describe exactly all the methodologies used to define the didactical contents as well as the theoretical bases on which the entire program is founded.

Why was the Weekly Food Plane before (TO) and after (T1) not used in the control group?

The weekly food plan was a specific activity of the MNP that allowed us to measure the improvement in the students’ skills only within the MN group. Each student from MN group was asked to compile a Weekly Food Plan, both before starting any activity related to nutrition issues (T0) and at the end of the didactic path (T1). In this way each student had his/her own starting level (T0) and any improvements due to the performing of the program was compared to the basic level. In addition, activities similar to the Weekly Food Plan are not typically carried out in the curricular program that represented the standard intervention for the control group. However, this represents a limitation of the study and other studies will be carried out to substantiate that MNP improved nutritional skills. All these considerations are now in the text (lines 97-99; 108-124).

In lines 97-99 there is information on the approval of the ethics committee

The modified text was marked in yellow (lines 118-121)

Was the level of knowledge assessed at the beginning, that is, before the start of nutrition education in both groups?

We thank the reviewer for this questions that allowed us to improve and clarify this aspect. The design of the study was organized as a case-control study (MN, the case, and CO, the control group). To MN group was administered the MNP didactic path through the web platform www.maestranatura.org: power point presentation, exercises, evaluation tests, experiments, practical activities and recipes to make at school with teachers and at home with parents. The didactic intervention of MNP spanned the entire scholastic year; CO group got one frontal 2h-lesson by an expert nutritionist focused on food groups, different meaning of food and nutrients, and the significance of Food Pyramid plus curricular lessons (generally not more than two 1h-lesson) about scientific topics held by the teacher. The level of the acquired knowledge was measured and compared at the end of the respective interventions, within one week from the end of the activities (text modified accordingly, lines 97-99; 111-114)

In lines 97-99 there is information on the approval of the ethics committee. In my opinion the groups a MN group and a CO group should have been organized according to the level of knowledge at the beginning of the study. Organizing them according the socio-cultural and economic characteristics and lack of measurement of nutritional knowledge at the beginning assumes that the knowledge of both groups is the same. Such an assumption is wrong.  Comparison of knowledge/skills between both groups at the end is of little value in such a situation.

We agree with the reviewer that this aspect may represent a limitation of the study. In the new revised version we pointed out that the lack of knowledge assessment before to carry out the didactical interventions, both in MN and in CO groups, represents a limitation of the study and the need to corroborate the present results by further studies to elucidate this aspect. This consideration was included in the limitations of the study (lines 262-266, marked in yellow)

However, we can add as further element strongly suggesting a substantial homogeneity between the two groups that all the students were asked to compile a validated questionnaire (KIDMED questionnaire) to check the dietary habits that resulted completely comparable in the two groups.

Finally, we were in some way obliged to skip the pre-evaluation test by the teachers that judged not friendly and seriously at risk of eliciting a feeling of inadequacy in the students requested to answer to questions, some of them quite ‘technical’, that they could not know without having before proper teachings

The characteristics of the study group are not presented.

In each school, a MN group and a CO group were organised so that the socio-cultural and economic characteristics of the students of the two groups were as homogeneous as possible. We added a sentence to the text to clarify this point (lines 84-86).

The information “In each school, a MN group and a CO group were organized so that the socio-cultural and economic characteristics of the students were as homogeneous as possible” is not presented on Figure 1. The results are presented for about 80% of groups presented on Figure 1. This information should be added on Figure, especially when the test was made only on the end stage of study.

The requested info is now included in the figure 1

Line 171-174 Is this part of the table?

Yes, it is the footnote of the Table, that, however, was shortened as suggested by the reviewer 2

This information is still too long

We further shortened the information